# Neurocircuitry of Personality Traits and Intent in Decision-Making

**DOI:** 10.3390/bs13050351

**Published:** 2023-04-22

**Authors:** Felippe Toledo, Fraser Carson

**Affiliations:** 1Department of Physiotherapy, LUNEX International University of Health, Exercise and Sports, L-4671 Differdange, Luxembourg; felippe.toledo@lunex-university.net; 2Luxembourg Health and Sport Sciences Research Institute A.S.B.L., L-4671 Differdange, Luxembourg; 3Department of Sport and Exercise Science, LUNEX International University of Health, Exercise and Sports, L-4671 Differdange, Luxembourg

**Keywords:** personality, mPFC, neuroimaging, neurocircuitry

## Abstract

Even though most personality features are moderately stable throughout life, changes can be observed, which influence one’s behavioral patterns. A variety of subjective assessments can be performed to track these changes; however, the subjective characteristic of these assessments may lead to questions about intentions and values. The use of neuroimaging techniques may aid the investigation of personality traits through a more objective lens, overcoming the barriers imposed by confounders. Here, neurocircuits associated with changes in personality domains were investigated to address this issue. Cortical systems involved in traits such as extraversion and neuroticism were found to share multiple components, as did traits of agreeableness and conscientiousness, with these four features revolving around the activation and structural integrity of the medial prefrontal cortex (mPFC). The attribute of openness appears scattered throughout cortical and subcortical regions, being discussed here as a possible reflection of intent, at the same time modulating and being governed by other traits. Insights on how systems operate on personality may increase comprehension on factors acting on the evolution, development, and consolidation of personality traits through life, as in neurocognitive disorders.

## 1. Introduction

There has been increasing interest in further understanding how the maturing of neural circuits throughout life affects the development of one’s set of behavioral traits. Constructing and consolidating self-identity is described as an ongoing search for values and commitments, which will guide decision making and the consequent attitudes in all life domains [1,2]. As the complexity of the matter of identity easily extends beyond the confinements of psychology, infiltrating the multiple fields of thought and research, such as philosophy, theology, and sociology [3], one may effortlessly lose track when discussing the subject matter of identity. To avoid falling into the many pitfalls the investigation of this topic offers along the way, individual personality traits offer an alternative for argumentation and measurement, often clearly recognizable and partially described in behavioral studies, such as control adverse behavior and consideration for future consequence (CFC). While individuals with control adverse behavior tend to act oppositional towards exogenous control over one’s freedom of choice [4,5], often leading to impulsive actions, individuals engaging in CFC are more prone to developing concerns about future repercussions, often finding themselves on the other side of this spectrum, where overly cautious conduct can lead to low daily discouraging rates [6].

In order to understand these behaviors at a neural circuitry level, these actions can be categorized into predefined personality traits. The currently most accepted model describing personality traits, the big five [7], and later developed by other authors [8], describes personality across five dimensions: extraversion, neuroticism, agreeableness, conscientiousness, and openness to experience. Although other models can be found in the literature, the elements of personality described in the “big five model” have been shown to be highly useful in analyses correlating personality and general health aspects [9]. One of the reasons for the success of this model is the connection between its features and the concept of identity, as an outright stable, but also fluid, self-construct [3,9].

Although most features of personality traits appear to maintain a reliable stability from 18 years old into adulthood, with small progressive decreases in neuroticism, extraversion and openness, and variable minor changes in agreeableness and conscientiousness [9], over the course of development, adolescents may experience higher fluctuations in behavior than the average adult [2,10,11]. These fluctuations can be perceived as a form of instability, but in most cases, do not represent a pathological state. The experiences leading to these fluctuations are often marked by a phenomenon known as identity uncertainty, a process defined by the ongoing reconsideration of commitments, associated with difficulties in social acclimatization and the risk of developing mental health disorders, such as depression [11].

Identifying the neurobiological mechanisms behind the above-mentioned behavioral diversity, modulating the expression of personality traits often reflected in one’s decisions can, therefore, be helpful to further comprehend the endogenous and exogenous factors influencing behavior. Current neuroimaging evidence, in particular of structural studies, may provide an opportunity to review behavioral theories in contrast with physical evidence. The neural circuits underlying social behavior in these cases can be observed by investigating how morphologic properties of pre-determined regions of interest (ROIs) inside encephalic circuits relate to one another [12]; a concept known as structural covariance. A peak in global covariance can be observed between the ages of 9.5 and 14.5 years old, with findings indicating that the transition from childhood to adolescence is marked by essential circuitry changes [13,14] that are noticeably reflected in the individual’s behavior.

Hence, this narrative review aims to explore the relationships between brain development and core elements associated with personality traits contributing to the maturing of behavior, observed in one’s judgement. In the pursuit of this goal, we investigate structural covariances established across the literature, involving circuits associated with decision making and social behavior, summarizing results of case–control and single-arm studies pointing to neural circuits acting on decision making in an assumed consistent matter, identified by the five categories of personality. Here, we present parallels between experimental and observational studies to draw attention to these converging points, and at the same time, commenting on differences and similarities in research methodology.

## 2. Literature Search

The National Library of Medicine (PubMed) was used for the acquisition of information used in this review. The search was performed between September 2022 and January 2023, focusing mainly on longitudinal and single-arm studies. Studies investigating any form of intervention were excluded, as well as studies where population diverseness could create bias on behavioral or cognitive assessments. Studies applying solely other non-invasive neuroimaging methods rather than structural or functional MRI were likewise excluded.

The initial search resulted in 16 longitudinal studies selected for the initial development of this paper. Among those, three studies were conducted as secondary analysis, utilizing data from the Human Connectome Project (HCP); seven of those were single-arm studies, one case–control and one longitudinal study. A total of 3604 patients were examined in these studies, with the most common structural assessment based on the acquisition of T1-weighted images, at 3 Tesla—only one study used 1.5 Tesla. Variety in the software used for analysis is present, but no significant differences between the systems is present. Although different cognitive and behavioral assessments were used, the majority of studies performed the NEO five-factor inventory (NEO-FFI) or a similar assessment for personality traits. The summary of proceedings and findings can be found in Table 1.

Additionally, four review papers were identified on a secondary search—focusing on reviews—for the purpose of providing commentary on the experimental studies and identifying potentially contradictory evidence in previous studies (gray literature search). The results of this secondary search added to the initially selected studies comprised a total of 24 articles, which suited the scope of this paper, on macroscopic and/or functional changes in neural systems related to personality traits across life.

## 3. Extraversion and Neuroticism

Extraversion and neuroticism are two prominent traits to be considered when discussing personality traits and decision making, as they may reflect neurofunctional changes responsible for affective and anxiety-related shifts throughout lifespan [9,15,16], directly affecting decision making. Extraversion can be understood as a likelihood of an individual to pursue pleasurable prompts from current events, as the tendency to novelty-seeking, and social engagement [9]. Behavior guided by extraversion may come across as highly motivated, adventurous and embracing of new experiences. Individuals guided by this trait are often observed taking risks that at times may even lead to hazard. This impact in decision making, creating a behavioral pattern over time, allows an external observer to make judgement of one’s reliability and risk-taking expectancy.

In parallel, neuroticism is best described by the enhanced responsiveness and expectation related to negatively charged environmental cues, as the perceived stressors found in social interactions, leading to an individual’s exacerbated threat assessment [8,9,17]. Neuroticism can be seen, at least in terms of decision making, as the antagonistic force to extraversion, in a way that leads to avoidance of risks, overvalued by what is perceived as eminent threat, refraining the individual from engaging in exploration. The balance between risk taking and avoidance is the result of equitable forces of neuroticism and extraversion, acting as counterpoints in the neurocircuitry melody of decision making.

These antagonistic influences of extraversion and neuroticism traits can be seen in a variety of neuroimaging studies [9,18,19,20]. A fitting example of this relationship between neurobiological markers or characteristics, and aspects of personality traits can be found in the myelination levels on prefrontal regions, which can be traced back to neuroticism, agreeableness, conscientiousness and extraversion [21]. In the literature, among the most common regions of interest (ROIs) for neuroimaging studies are the anterior cingulate cortex (aCC), medial or dorsomedial prefrontal cortex (mPFC), the dorsolateral prefrontal cortex (dlPFC), and the ventromedial prefrontal cortex (vmPFC) or orbitofrontal cortex (OFC), with authors often including both more medial and more lateral (ventrolateral prefrontal cortex—vlPFC) parts when using this description. All these areas of the PFC are associated with executive functioning, playing a key role in decision-making processes, acting on behavioral strategies and patterns, through allocentric and egocentric information processing [18,22,23].

The mPCF, in a circuit involving the dlPFC and the posterior cingulate cortex (pCC) is seen acting on decision making in self-regulatory processes of emotional value, with the purpose of identifying, processing and deciding on the self-relevance of exogenous stimulation [2]. This ability to define “if” and “to which extent” stimuli are to be considered relevant enough for decision making, will impact social and adaptive behavioral traits of another circuit involving the dmPFC and premotor cortex (PMC). This adjacent circuit carries attributes of egocentricity and self-absorption, associated with the decreased activation of the PMC [24], possibly restraining “self-relevancy” of actions to isolated individual interests, rather than overall benefits to the social group the individual plays a part in. Similarly, decreased activation of the mPFC in this circuit would contribute to impulsivity and carelessness [24], obscuring or fully eliminating one’s ability to reflect on future consequences of actions, both for the individual and the social group. Furthermore, the surface area of the mPFC seems to be of particular relevance in neuroticism, with the increase being directly associated with levels of neuroticism in young adults, with evidence pointing to the right mPFC playing a stronger role in this trait than the left, while other associations appear as less relevant for this trait [18,19].

When addressing behavioral changes in the aging process through the lens of neuroanatomical and neurofunctional changes over time, variation in dmPFC CTh, particularly on the left-brain hemisphere, offers another opportunity to verify the issue of neuroticism and extraversion as two sides of the same coin. In healthy elderly participants (mean age 70, 3, SD = 6,6), increased bilateral CTh on the mPFC appears to be related with increased levels of extraversion, while decreased CTh on the same region is usually associated with neuroticism [9]. A previous study had already presented similar results [25], but with significant discrepancies between right- and left-brain hemispheres, implying the role of the left dmPFC alone to be responsible for the levels of extraversion and neuroticism. Another study was able to show a decrease in cognitive flexibility associated with age, linked to decreased OFC CTh, especially closer to the midline, which is thought to additionally impair reversal learning [26], while more recent studies imply a negative relationship between cognitive flexibility and neuroticism simultaneously to a positive relationship between cognitive flexibility and extraversion [27], posing again these two personality traits as counterparts, and at the same time, demonstrating the strong link between them and the structure and function of the OFC in decision-making processes.

One could argue that even though impulsivity has been mostly associated with higher levels of neuroticism, carelessness, and impetuous behavior, it can also be observed in frontal syndromes resulting from acquired brain injury, with extraversion becoming a more prominent personality trait. The question here would probably be better addressed in the domain of intent, guiding careless actions based on fear-related issues or disregard for future consequences. Attending to intent as the subject matter behind decision making would be helpful in order to expand the view on neurocircuitry. For this purpose, the OFC may offer a more precise understanding, due to its role in fear extinction and adaptive learning. An inverse relationship is found between cortical thickness (CTh) on this area and increased levels of neuroticism in young adults [9,28], with a continuous physiological CTh decrease among mPFC throughout the course of life, starting in adolescence [2,29].

Additional evidence supporting the idea of fear-based impulsiveness and lack of consideration for future consequences linked to neuroticism can be found in the observation of acquired brain injury patients, where damage to the OFC is linked to behavior guided by immediate outcomes [6,30,31,32]. As seen above, the mPFC can also contribute to the aspect of neglecting to reason about future consequences [18,19,24], although not necessarily implying that aggressivity would be a definite trait. Perhaps one important determinant of whether an impulsive action would carry traits of aggressivity would be the matter of threat recognition. The circuit OFC-aCC-aINS-AM appears akin to increase activation during passive viewing of aversive stimuli, contributing to one’s ability to appraise events as hazardous or not, even in the absence of triggers [33]. The ability to reverse learned fear responses, or fear extinction associated with this system has proven to be of great value in understanding the impact of anxiety-related and posttraumatic mental health issues affecting the presence of somatic markers, crucial for understanding fear learning and extinction [6,30,34].

In the subject of neurobiological markers associated with aggressive behavior, attention must be granted to the left dlPFC. This region already plays a role in the matter of consideration for future consequences and impulsive behavior, and appears highly relevant in negative affect, leading to aggressivity, mostly in males, but consistently across studies [20]. Aggressivity, despite the caring components of conscientiousness and extraversion, has mostly been connected to increased levels of neuroticism [35,36], reinforcing the link previously discussed between aggressivity, impulsivity, and threat perception. The aCC in a circuit compelling the aINS and substantia nigra (SN) will also leave its imprints in decision-making processes stirring social behavior. Here, the aCC will play a role in the maintenance of the cognitive sets required for a task or interaction, influencing reaction to exogenous influences on one’s freedom of choice [4,5,37].

## 4. Agreeableness and Conscientiousness

Similar to extraversion, agreeableness, the tendency of a person to be collegial and trusting [38], is primarily seen as a function of interpersonal behavior, with the contrast that it describes the quality of the relationship between extremes of compassion and antagonism [8]. In this way, agreeableness can play a direct role in how one’s self-image may affect social attitudes, through trust, straightforwardness, altruism, compliance, humility, and tendermindedness [8]. The concept of conscientiousness appeared in the literature for the first time in a description by Hartshorn et al. (1929), as a reflection of ego strength, affecting willpower, initiative and social responsibility, supporting the link between mindset and action. This personality trait has a direct influence on behavioral features, such as dependability, orderliness, and achievement-oriented actions, also described as goal-directed behavior [39]. Therefore, agreeableness and conscientiousness seem to be interwoven traits of decision making through which one ponders about social responsibility, liability, and accountability, with a seemingly more “compassionate” individual demonstrating more will power to act in favor of others in a social context, while an antagonist behavioral pattern might lead to lack of initiative to engage in communal needs.

The matter of intention also requires special attention here. It is not the purpose of this paper to discuss ethical implications of social actions; however, one cannot examine the subject of individual responsibility (or the perception of it) in an organized society without mentioning intent and perceived social image. Studies investigating these aspects have found increased mPFC connectivity in young adults when participants were observed while evaluating others, as well as themselves [40,41,42,43], in accordance with studies on self-conscious emotions affecting decision making [44,45,46]. These results suggested that increased mPFC connectivity might be a trait that reflects one’s wishes to conform with socially acceptable behavior, with the intention to belong to a group “sheathing” rationale.

This phenomenon is not new to behavioral sciences. Its marks can be found in different theories, such as fallacy (argumentum ad populum) or cognitive bias (bandwagon effect), describing an individual favoring socially desirable behavior. In these instances, where “expected” behavior, or what the individual believes to be the desirable behavior determined by peers, an increase in mPFC connectivity can also be observed [47,48,49]. However, to make a conscious decision in favor of expected social behavior, one must often consciously or preconsciously be able to drive the focus of a thought path towards it, inhibiting or suppressing other assertions. This is where the role of the aCC in determining the focus of attention becomes essential, and can be seen in neuroimaging studies, in the form of decreased aCC myelination [50].

A similar region of interest is the OFC, particularly the more medial parts, often referred to as vmPFC, which seems to also impact decision making in social contexts, but through more specific mechanisms. Activity on the OFC has been linked to self-enhancement (evaluating oneself above average) and self-serving behavior in situations aiming to increase one’s own value in a social group [40,51,52]. This participation of the OFC in behavioral traits can be confirmed by pathological studies, where acquired injury to the OFC and aINS in adults was associated with reduced self-conscious emotions and engagement of these patients in what would be considered socially inappropriate or undesirable behavior [40,53,54].

Once more the matter of intent can be used to elaborate on the link between neuroimaging results and observed behavior. The circuit mPFC-dorso-rostral aCC-pCC-PRECUN has been associated with introspective processes for task-induced appraisal of relevance (Schmitz and Johnson, 2007) and self-representation [2]. With the addition to the dlPFC in the circuit mPFC-dlPFC-pCC, the aspects of self-relevance and emotional regulation [2] infiltrate decision making. The implication of these interactions is that the observed behavior becomes a response to a specific social situation, making its nature (confirmative or dissenting in intention) indistinguishable for an external observer. If this is the case, the argument that studies may, at times, also investigate motives for decisions made, would be difficult to sustain here, since the participant’s answer to these questions would also be biased by the same systems that created the response in the first place.

This matter becomes more evident, or at least slightly clearer, when subcortical systems are brought into the equation. Connections between prefrontal areas, with emphases on midline regions (i.e., vmPFC and mPFC) and the basal nuclei might be useful to delineate intent and expectation of outcomes. Increased functional connectivity between the PFC and the nucleus accumbens (NAcc) has been observed during the pursue of self-defined goals [55,56], as motivation and affect [57]. Furthermore, connectivity between the vmPFC, INS, caudate and NAcc has been observed in prospective reasoning, or anticipation of future outcomes of decision, with reduced INS CTh being observed in individuals presenting reduced levels of empathy [33,58]. A completion of this circuit in the particular aspect of one’s valuing of communal needs can be accomplished by adding the AM and aCC to this equation, with the dorsal-aCC playing an import role in controlling adverse behavior and resistance to framing [4,59]. These two ROIs have been linked to the identification of goals and behaviors aiming to serve self-well-being in a social group—often linked to survival in hominids and other social animals [33].

## 5. Openness

Elaborating on the matter of intent and openness, among the five personality traits, presents a somewhat unique opportunity. Defined as the attitude to be overt to experiences, openness may be subjected to influences of neuroticism driven by fear, extraversion driven by impulsivity, agreeableness driven by willingness, and conscientiousness driven by focus. Individuals engaging with this trait are often described as open-minded and less rigid in their reasoning. In this sense, openness can perhaps be viewed here as the first filter in the frame of intentions, where an individual would or would not allow for the possibility to engage with new experiences before willingness and focus set and a decision for (or against) this engagement is made.

As previously mentioned, openness is associated with myelination rates in the mPFC, aCC, pCC, pINS and PUT [21]. In males, high levels of openness suggest increased OFC-aCC activity, while in females, the most anterior portion of this circuit appears to have its main “nodes” located in the frontopolar cortex [60,61]. Despite this solely somewhat relevant difference between the sexes, the medial structures of the frontal lobe seem to rule over the morphological aspects of this personality trait.

However, despite increased connectivity in frontal midline “hubs” playing an important role in modulating levels of openness, connections between the aCC, dlPFC and PRECUN have also been associated with increased levels of openness [61,62], as has the connectivity between SN-VTA-dlPFC in the mesolimbic system [61,63], related to goal-directed behavior. This result often present a challenge in research, with openness appearing to be more of a direct result of exhaustive complex reasoning processes, with components spread throughout different circuits, depending on the situational context the individual is a part of.

Among the features associated with openness, creativity might be one of the most researched. From previous studies on sociocultural tendencies conducted in the 1950’s to more recent neuroimaging studies [64], creativity has been long associated with marks of social intelligence and academic achievement [65,66,67], as well as milestones in developmental sciences. Creativity has been widely associated with the diversifications of the prefrontal systems previously mentioned, being linked to lower left OFC CTh, along with lower raCC volumes [68]. Despite other studies not verifying these results [43,69], they still support the premise that cortical thinning, as an ongoing physiological process throughout life, is connected to higher circuitry efficiency at lower neuroplastic demands. This phenomenon of decreased volumes leading to higher efficiency has also been investigated in other areas associated with creativity, and therefore, openness. Artistic creativity has been associated with decreased bilateral volumes in the psPC and PRECUN [20], left AM, and left caudate [69].

## 6. Limitations

One limitation of this narrative review is the focus on gray matter changes. Although more research has been conducted on this aspect, advances in neuroimaging, especially in tractography, over the last five to ten years have been providing extremely valuable information of brain function. Further investigation, either in single-arm studies or in longitudinal frames, would likely clarify many of the questions and hypotheses posed here. Understanding the effects of strengthening and degradation of white matter tracts connected to “hubs”, such as the mPFC, aCC and psPL, as well as limbic structures, could aid greatly in comprehending how behavior is modelled across life, as well as how physiological changes associated with aging contribute to decision making.

Another point worth mentioning is the paradigm adopted for the frame of this work. The choice of investigating personality traits, instead of the development of identity or even personality as a whole was made due to the number of different ideals of these two concepts. For the comparison with structural and functional MRI data, this paper strictly followed the theory of traits, circumscribed by objective tests, many of them could also be performed during neuroimaging procedures. This means that instead of observing personality by considering all its influences (environmental, genetic, socio-cultural, etc.), only the five traits were taken into consideration in a restricted set. Furthermore, the methodological choice for a narrative, rather than other forms of review, definitely limits the results found to one facet of the many possible approaches, with other streams probably leading to divergent results. This choice was also a conscious one, focusing on a few aspects, rather than solely superficially approaching multiple points.

## 7. Conclusions and Future Directions

The aim of this narrative review was to investigate the relationships between the development of neurological circuits and personality traits that impact decision-making processes. To address this goal, the circuits associated with decision making and social behavior in the previously mentioned ROIs have been formerly discussed in the subject of extraversion, neuroticism, agreeableness, and conscientiousness. On the one hand, the observed overlapping of systems may appear obvious with the role of these areas in behavior being extensively presented throughout the literature. However, although logic dictates that these systems should overlap in a neuroarchitectonic sense, it is also noticeable that openness, in contrast to other personality traits, almost solely overlaps with the frontal portion of these circuits. This facet could simply indicate that more focus on research for biological markers for openness than other traits in needed, or it could signify that when openness becomes the target of neuroimaging studies, the subject of intent transcends from a philosophical to a “palpable” question.

The inference of this review is that openness may represent the subject of intent, as in one’s conscious, preconscious, or even unconscious, resolving to engage in neurotic, extraversive, compliant, or mindful actions towards oneself and others, susceptible to the perceived “rewarding consequences” of the behavior in question. This could explain why the issue of neuroanatomical correlates for openness appears more controversial in the literature than other traits, with authors reporting on findings, especially regarding CTh [61,70], while others struggle to identify absolute correlations [71]. The question of whether this hypothesis is reasonable or the difficulties in finding coherent neurocorrelates to openness is simply a matter of methodological (or technological) challenges should also be considered here.

Despite the mPFC being extremely frequent in the literature, investigating reasoning in trait judgement [72], reasoning, knowledge, and situational awareness [73], its specific role or isolated function is still highly disputed [40]. The matter that this region acts as an integrative area for reasoning linking other portions of the brain and simultaneously developing rationale itself, likely indicates that the mPFC is the ground on each reasoning of argument itself. Given the depth of the topic of personality in goal-directed behavior and social responsibility, one should humbly expect to encounter barriers, more often than not infringing the borders of philosophy, sociology, anthropology, and other disciplines, each with a unique, but at the same time, complementary view on the matter.

Despite this well-known impediment, two statements can be made at this point: (1) The circuits govern personality converge at a common point around the mPFC; and (2) Neuroimaging methods can facilitate the understanding of behavioral assessments by providing more specific evidence on the reason behind choices. Understanding how different systems integrate to produce the outcome of one’s personality traits, could increase comprehension on the development of identity throughout life. In particular, the changes individuals undergo through adolescence and advanced age, but also potentially supporting the diagnosis and treatment of frontal and disconnected syndromes.

## Figures and Tables

**Table 1 behavsci-13-00351-t001:** Summary of the findings.

Author	Population	Design and Measurements	Main Findings *
Becht et al. [2]	160 right-handed adolescents (mean age = 15.92 ± 2.97) 54% female	Longitudinal study 3 assessment waves separated by 1.3-year interval, over the course of 4 years T1 structural imaging (3Tesla) Analysis software: FreeSurfer^®^	Adolescents in the identity moratorium showed a more positive linear slope than comparison group, indicating delayed maturation and higher NAcc volume levels across assessment waves.
Keil et al. [15]	11 right-handed adolescents (mean age = 18.1) 54.42% female	Single-arm study T1 structural imaging (3Tesla) Analysis software: not mentioned EEG, 267 electrodes, impedance below 50 kΩ 0.1 Hz high-pass and 100 Hz low-pass	Increased coupling for arousing content occurs for large-scale electrical activity at a cortical level, with cortical regions (right PRECUN, bilateral inferior, and superior posterior parietal lobes, left mTG, left iTG, left post-central cortex, right iFG, and right mFG), consistent with regions targeted as elements in a functional network.
Massey et al. [16]	41 individuals with schizophrenia (mean age = 31.79 ± 8.56) and 46 healthy controls (mean age = 32.91 ± 6.69) 47.8% female (schizophrenia group) and 34.4% female (control group)	Case-controlled study T1 structural imaging (3Tesla) Analysis software: FreeSurfer^®^ Emotional Perspective taking Task (EPT) Interpersonal Reactivity Index (IRI) Questionnaire of Cognitive and Affective Empathy (QCAE)	Individuals with schizophrenia presented deficits in cognitive empathy, associated with lower cortical thickness in iFG, INS, right SMA and TPJ, bilateral mPFC and amCC.
Miskovic et al. [17]	1033 inmates (mean age = 31.80 ± 7.05) Only male	Single-arm study T1 structural imaging (3Tesla) Analysis software: FreeSurfer^®^ Psychopathy Checklist Revised (PCL-R) Addiction severity index (ASI)	Psychopathy scores were associated with reduced gyrification in the right psPL, dlPFC, and mCC, as well as reduced functional connectivity between mCC and right psPL.
Roman et al. [18]	283 participants (mean age = 23.32 ± 5.07) 51.94% female	Single-arm study T1 structural imaging (3Tesla) Analysis software: Surfstat^®^ Adult decision making competence (A-DMC) Vocabulary subscale of Shipley 2 battery—crystallized intelligence LSAT—logical reasoning	Decision-making competence showed association with logical reasoning, linked to individual cortical surface area differences in daCC and right psTS.
Rudorf et al. [4]	50 students (mean age = 22 ± 4) 56% female	Single-arm study T2 weighted image with BOLD contrast (3Tesla) Analysis software: FSL^®^ Control aversion task	SN connectivity was associated with individual behavioral tendencies for cooperation with peak locations being piPL, psPL, postcentral gyrus, mFG, OFC, INS, aCC, PUT, iFG, pCC, PRECUN and mPFC.
Cao and Xia [6]	203 adults (mean age = 20.23 ± 1.93) 51.23% female	Single-arm study T1 structural imaging (3Tesla) Analysis software: SPM8^®^ Consideration for Future Consequence Scale (CFC) NEO five-factor inventory (NEO-FFI)	Surface area of the vmPFC positively relates to most traits of conscientiousness, except for deliberation.
Sheng et al. [19]	19 adults (mean age = 27.7) 68.42% female	Single-arm study T2 weighted image with BOLD contrast (3Tesla) Analysis software: SPM2^®^ Psychopathy Personality Inventory Revised (PIP-R)	Carefree nonplanfulness subscale scores correlated with mPFC deactivation during task.
Toschi and Passamonti [20] **	1003 adults (mean age = 29) 54.83% female 83.6% right-handed	Secondary analysis T1 and T2 weighted Analysis software: FSL^®^ NEO five-factor inventory (NEO-FFI)	Neuroticism related positively to intracortical myelin in the visual cortex, and negatively to FPC myelin content. Extraversion related positively to myelin levels in the psPL, and negatively in the aCC. Aggreableness related positively to myelin levels in the OFC. Conscientiousness related positively to intracortical myelin content in the FPC and negatively in the daCC. The negative association between openness and myelin is spread throughout the telencephalon.
Vartanian et al. [21]	185 adults (mean age = 22.06 ± 3.6) 50.81% female	Secondary analysis T1 structural imaging (3Tesla) Analysis software: FreeSurfer^®^ Big Five Aspects Scale (BFAS)	Openness and intellect were negatively correlated with cortical thickness in rostral and superior prefrontal regions, and positively correlated with surface area and folding in the OFC, parietal and temporal areas.
Vijayakumar et al. [12]	192 children (between 9.5 and 14.5 years old)	Longitudinal study Three assessment waves separated by approximately 18-month intervals. Not all participants engaged in all 3 assessments, and a sliding window approach was used to create age bins. Parent report sexual maturity scale Wechsler Abbreviated Scale of Intelligence (WASI-II)	Transition from childhood to adolescence was characterized by global increases in structural covariance, followed by reductions in mid-adolescence. Developmental patterns in nodal and modular properties are consistent with earlier development of motor skills, with consistent refinement of emotion modulation, social cognition, and executive faculties.
Wertz et al. [22]	248 participants (mean age = 22.8 ± 3.5) 48.79% female	Single-arm study T1 structural imaging (3Tesla) Analysis software: FreeSurfer^®^ Creative Achievement Questionnaire (CAQ) Wechsler Abbreviated Scale of Intelligence (WASI-II)	Lower CTh on the left OFC and increased on the right angular gyrus were associated with CAQ scores. Results show that highly creative individuals exhibited cortical structure differences across bilateral sTG, isthmus cingulate, INS, psPL and SMA. Within artistic creativity measurements, authors describe inverse relationship to CTh in the left sTG-mTG, and right psPL. Higher artistic creativity was also related to decreased left AM volume and increased left CAU volume.
Wright et al. [9]	29 right-handed elderly participants (mean age = 70.3 ± 6.6) 58.62% female	Single-arm study T1 structural imaging (1.5Tesla) Analysis software: FreeSurfer^®^ NEO five-factor inventory (NEO-FFI)	PFC subregions were associated with extraversion (increased CTh of the right sFG and left mFG) or neuroticism (decreased CTh of the same regions) in elderly. In the right aTL, only neuroticism, was significantly associated with CTh.
Yasuno et al. [23]	37 adults (mean age = 28.1 ± 6.2) 27% female	Single-arm study T1 and T2 weighted imaging (3Tesla) Analysis software: SPM12^®^ NEO five-factor inventory (NEO-FFI) Minimental State Examination (at cutoff = 27 points)	Openness associated with myelination in the right aCC, mPFC, pCC, pINS, and PUT.
Yoon et al. [24]	43 adults (mean age = 23.7) Control group: mean age = 23.1; 47% female Observation group: mean age 24.3; 55% female	Case–control study fMRI, 2 × 2 × 2 mm voxel resolution	Confirmation of the observer role linked to increased mPFC and aINS activity in impression management. Increased OFC activity in self-enhancement and self-serving behavior.
Zhu et al. [25] **	1113 adults (mean age = 28.79 ± 3.71) 54.7% female	Secondary analysis T1 structural imaging (3Tesla) Analysis software: FreeSurfer^®^ Adult Self-Report (ASR) National Institutes of Health Toolbox (NIH toolbox) NEO five-factor inventory (NEO-FFI)	Relationship between aggressive behavior and negative affect with CTh in the dPFC.

* Relevant for this article. ** Data collected from the database of the Human Connectome Project (HPC). Data extraction table: NAcc = nucleus accumbens, PRECUN = precuneus, mTG = middle temporal gyrus, iTG = inferior temporal gyrus, iFG = inferior frontal gyrus, mFG = middle frontal gyrus, INS = insular cortex, SMA = supplementary motor area, TPJ = temporoparietal junction, mPFC = medial prefrontal cortex, aCC = anterior cingulate cortex, psPL = posterior superior parietal lobe, piPL = posterior inferior parietal lobe, mCC = middle cingulate cortex, psTS = posterior superior temporal sulcus, PUT = Putamen, vmPFC = ventromedial prefrontal cortex, FPC = frontopolar cortex, OFC = orbitofrontal cortex, sTG = superior temporal gyrus, AM = amygdala, CAU = caudate, sFG = superior frontal gyrus, pCC = posterior cingulate cortex, aTL = anterior temporal lobe/temporal pole (TP), dlPFC = dorsolateral prefrontal cortex.

## Data Availability

Not applicable.

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
