# Peer review of "Neurocircuitry of Personality Traits and Intent in Decision-Making"

_behavsci, 2023, doi:10.3390/bs13050351_

Round 1

Reviewer 1 Report

The review presented is quite interesting and offers a contribution to the studies of human behaviour, connecting psychological concepts - personality and identity -  with neuroscience. 

It would be important, however, that the authors present more information about the process of mapping and selection of the studies that were incorporated and, above all, about the characteristics of these studies: years of publication and methodological characteristics (samples, measures, procedures, types of analysis).

This information can be synthesized and presented in a summary table.

These characteristics can also be used to make notes on the scope and limits of the studies reviewed - as well as the scope of the considerations  in the manuscript under analysis.

Based on this, in the conclusions, the authors can indicate the importance of new studies and the methodological care that should be taken to increase the reach of the conclusions. 

Author Response

We would like to thank the reviewers for the comments and assistance in developing the manuscript. A full response to each point is provided in the attached 'Reviewer Response' document.

Reviewer 2 Report

The review of Toledo and Carson has the ambitious goal of characterizing neurophysiological circuits that are correlated with personality traits, as described by Big Five model. The authors begin by describing the various personality domains and then present the most recent neuroimaging evidence about the brain circuits that are associated with trait domains implicated in decision-making and social behavior.

The review is well-written and well-structured. However, I have a few suggestions for the authors that could further improve their manuscript.

The main strength of the manuscript is the detailed and clear description of personality traits from a theoretical perspective. However, when discussing neuroimaging studies, the authors should provide a more detailed description of the specific techniques used to obtain the results, as well as the experimental procedures in which the subjects were involved. For example, the authors should describe the methods used, such as fMRI or other techniques.

Additionally, it would be helpful to know whether the participants in the reviewed studies performed a specific task during the neuroimaging studies or were subjected to other types of experimental manipulations. This information could provide better context for the discussed neurophysiological results in relation to the assessed behavior.

Furthermore, when the authors report evidence from individual or social groups, as well as young or elderly subjects, they should inform the readers about the specific characteristics of the participants assessed in the reported studies.

In my opinion, the main weakness of this manuscript is the general conclusions that do not allow for clear correlation with the evidence in the literature. Therefore, I suggest that the authors include more accurate information about the results from different studies and how these results were obtained from specific experimental populations or experimental procedures.

Author Response

(The authors gave the same response as above.)

Reviewer 3 Report

The article presents an original view on the relationship between personality traits and neurophysiological activity of the brain. The authors make an attempt to find evidence of this relationship in research on decision-making and provide arguments in favor of this relationship. The data on the topographic correlation of brain regions with specific behavioral actions and personality manifestations are very interesting. 

However, the theoretical basis of the review looks questionable.

1.     There is terminological fuzziness (confusion) in the description of the basic concepts associated with personality. The authors rightly emphasize the connection between the maturing of neural circuits and personality development (line 28-29). But they designate the development of personality as “constructing and consolidating self-identity ”. These are not identical processes. Self-identity is only one of the aspects of personality development. Personal development includes the formation of personality traits, character, attitudes, life orientations, life goals, and much more.. In turn, self-identity influences decision-making and the formation of values and attitudes. But the opposite is also true. The appropriation of values and decision-making experience affects the individual self-identity. The authors feel the vulnerability of this position; therefore, further they propose to use the concept of an individual trait. It would be logical to immediately start with this term and strictly adhere to it.

2.     The relationship between the concept of identity and the five-factor model of traits (line 52-54) does not mean that one measures the other. A personality trait is a behavioral construct; identity is a subjective sense of self, self-perception. They have completely different content, differing as external and internal. Therefore, the phrase "Although most features of identity and personality appear to maintain a reliable stability from 18 years old into adulthood" (line 54) is incorrect. As the direct linking of identity to extraversion or neuroticism is incorrect too. These analogies lead the authors to logical errors. Thus, a reference to variable changes of personality traits (line 55), is followed by an example of identity experiences in adolescents (line 59), followed by an unexpected transition to risk mental health disorders, social acclimatization  and depression, and suddenly a conclusion about the neurobiological mechanisms behind identity variations (line 62), as well as exogenous and endogenous factors of behavior.Identity, personality traits, depression, getting upset, susceptibility, adaptation, and behavior can of course be compatible. But these are multi-level and multi-order phenomena. Therefore, this part of the article gives the impression of a chaotic argumentation, as if the authors are confused in terminology about personality. If the focus is on the five-factor model of personality traits, then in the introduction it is necessary to justify that the construct “personality trait” is being considered, it has an appropriate definition, is associated with behavior, has a neurophysiological basis described in many personality theories (see, for example, Allport, Cattell, Eysenck and others). It is not necessary not to introduce concepts that are not relevant in the content of the main part of the article. In the conclusion, the authors again state that the considered "core elements of personality that accompany the development of self-identity" (line 290). Again, these are different phenomena. Personality traits are not indicators of self-identity. This position of the authors makes us think about their competence in the field of personality psychology.  In the conclusion, the authors state that "neuroimaging methods can facilitate the understanding of behavioral assessments”  (line 319). This statement is true, but has nothing to do with self-identity.

3.     3. It is unclear how articles for review were selected. . There are general description rules for reviews and article selection procedures (eg PRISMA). It is not clear what keywords were searched for, what additional information was taken into account, in what volumes. Articles with what evidence base (sample, methods) were included or rejected.

4.     4. For the basic psychological concepts - personality traits, the five-factor model, which the authors have chosen for analysis themselves, is not always used. It is not clear why the classical universally recognized definitions of extraversion from the works of H. Eysenck or R.McCraeand P. Costa were not taken as a basis. The definition of openness is not given at all (Section 4). If the purpose of the article is to emphasize the connection between personality traits and neurocircuits, then the definitions adopted in the theories of personality traits or in the original five-factor model should be used.

5.     The opposition between extraversion and neuroticism is questionable. It is unclear why they are antagonistic concepts and have antagonist influences (line 105). These are two different traits that may well be combined with each other. If in neuroimaging studies they show opposite results, in behavioral studies they are considered as conjugated with each other. So, the conclusion that " the balance between risk-taking and avoidance, would be the result of equitable forces of neuroticism and extraversion, acting as counterpoints in the neurocircuitry melody of decision making" (line 104) seems to be insufficiently substantiated and evidentiary. Just as unfounded are conclusions at the level of assumptions that " in frontal syndromes resulting from acquired brain injury, with extraversion becoming a more prominent personality trait" (line 152-153). A trait is a behavioral pattern. If we are talking about behavior (and its changes) precisely as a characterological pattern, then I would like to see specific examples that illustrate this, including just personality changes. The authors argue that “fear-based impulsiveness and lack of consideration for future consequences linked to neuroticism” (line 162-163) is a change in personality. What is the evidence that this is indeed the case? Maybe we are talking about mental disorders or attention deficits, which a person can just compensate for at the expense of other resources?

6.     For such an article, I would like to see a theoretical model that describes the relationship between the five personality traits, decision-making and intentions, as well as the neurocircuits or neuronal localization that provide them. The enumeration of disparate facts does not allow building a common understanding of this connection. As a result, the text of the article is a set of little-related research results of other authors, which only in the most general form support the idea of the article. The main content is that individual personality characteristics (and often not traits) can be interconnected. For example, decision making with self-assertion (line 225), evaluating others with emotions of self-awareness (line 209), evaluating relevance with self-presentation (line 235), intentions with expectations of results (line 246), evaluating oneself with selfish behavior (line 227). One would like to ask where is the agreeableness from the five-factor model? These examples are taken from section 3 on agreeableness and conscientiousness. But in other sections, examples are given that are scattered results of other people's research too, At the same time, it is absolutely unclear how they were obtained and for what research purpose, what limitations they have.

Author Response

(The authors gave the same response as above.)

Round 2

Reviewer 3 Report

Thank you for adding information about the procedure for finding review articles. This is an important addition that explains some of the author's judgments in the manuscript.

However, the main terminological issue in the manuscript remained unresolved.

The authors rely on the behavioral approach and the Theory of Traits. Their choice is theirs. This point of view takes place in the Personality Psychology. But the term “identity” or “self- identity” is from a different psychological paradigm.. The text of the article would look more convincing if the authors would strictly adhere to the theory of traits, and specifically the theoretical and methodological paradigm of the Big Five.

However, it should be recognized that the authors use one of the points of view. But even if we take into account the reliance on a five-factor structure, it is not clear why the term self-identity should be used. Most researchers recognize that a person is a system, that account for consistent patterns of feelings, thinking, and behaving (Pervin, Cervone & John, 2005, p. 6). They share in common the view that (a) personality is a psychological system, (b) composed of a group of parts (c) that interact, (d) and develop, and (e) that impact a person's behavioral expression. (Mayer, 2007) So personality cannot be described solely as a collection of traits. Complex integral formations such as identity, authenticity, self are the result of the interaction of biological, environmental and psychological factors. Therefore, the assertion that basic personality traits accompany the development of self-identification is very superficial.

Personality traits matter for self-identity. But they are also important for the formation of stable forms of behavior (which is not self-identification), and for assessing the characteristics of the response, and for self-education, and for the development of the self-concept and for authenticity, etc. All these are different personal phenomena, but they are not are mentioned in the text.

My persistent advice. Start with the concept  "a trait" and use only that throughout the manuscript. Do not mislead readers with confusing terminology and introducing concepts that are not relevant in the five-factor structure of personality.

I'll try to explain again. In the introduction and conclusion (conclusions and discussion), two concepts “self-identity” and “personality traits” are formulated. The self-identity dynamics and the change in personality traits is a well-recognised fact. But the problem is different. Self-identity in your article is not considered in any way and is not supported by neurophysiological research. Your manuscript on personality traits and the neurocircuits that support them. Self-identity has nothing to do with it. Its mention is not appropriate, since it is a subjective sociocultural phenomenon. Moreover, the answer to your 3rd comment only strengthens this statement. Your manuscript is about the behavioral phenomenology of personality.

Thank you for clarifying the new evidence for the content of the five basic personality traits, such as “openness is the act of being open to [new] experiences”. But then it requires an explanation in the text. If the original meaning that H. Eysenck or R.McCraeand P. Costaputintothe basic concepts have changed, then it is worth clarifying this. It should be clarified the statement that the definitions you used are more relevant to the modern sense of the specific trait.

Although the authors clarify in their answers, that the paper was constructed as a narrative review, not as a systematic review and not a meta-analysis. Some pretty significant conclusions follow from it. We would like the authors to once again clarify the limitations of the review.

We would like the authors to once again clarify the limitations of the review.

The limitation information that has been added applies only to neurophysiological studies. It should be added that the limitations of the article are also determined by the choice of the psychological paradigm (view of personality) in the description of personality. Trait theory is not the only concept, though it is quite common. However, within the framework of a different concept (for example, the H. Eysenck three-factor model or the 16-factor R. Cattell model, which also belong to the dispositional paradigm), we might have a different result. Not to mention other, no less common Personality Theories, developed in behavioral, humanistic, psychodynamic and other approaches.

Author Response

We would like to thank the reviewer for their comments and assistance in developing the manuscript. To improve the clarity and easy of understanding, we have removed reference to the terms 'identity' and 'self-identity' within the manuscript and have now focused solely on 'personality traits'. 

Comments on the section “limitations” were added, to emphasize the point requested here, that in the methodology chosen for this paper, not every aspect of each concept is being taken into consideration and that different streams of thought could lead to divergent results, as requested.

Further clarification of the term 'openness' was presented [line 307] to increase transparency on the meaning of the term and how it is being used in the context of this analysis.

Round 3

Reviewer 3 Report

Thank you. My comments were taken into account by the authors of the manuscript. No more comments